# Overcoming structural violence through community-based safe-spaces: Qualitative insights from young women on oral HIV pre-exposure prophylaxis (PrEP) in Kisumu, Kenya

**Patrick Mbullo Owuor**[1,2*], **Silvia Achieng Odhiambo**[2,3], **Wicklife Odhiambo Orero**[2,4], **Judith Atieno Owuor**[2,4], **Elizabeth Opiyo Onyango**[3]

**1** Departments of Anthropology and Public Health, Wayne State University, Detroit, Michigan, United States of America, **2** Pamoja Community-Based Organization, Kisumu, Kenya, **3** College of Health Sciences, School of Public Health, University of Alberta, Edmonton, Canada, **4** Department of Social and Development Studies, Mount Kenya University, Thika, Kenya

\* owuor@wayne.edu

## Abstract

Biomedical and behavioral interventions have led to significant success in the prevention of HIV/AIDS. However, in rural communities, structural violence persists and continues to create barriers to the uptake and utilization of health services, especially among young women. To overcome these barriers, community-led initiatives have provided a range of interventions, including safe spaces (i.e., vetted meeting venues where girls come together to discuss issues affecting their wellbeing and access health services, such as PreP) for young women. Although these spaces provide a safe haven for at-risk girls and young women, the role of community safe spaces in overcoming structural violence remains under-explored in literature. Using the structural violence framework, this study explored how community-led safe spaces for HIV prevention programs can overcome structural forces – policies, norms, or practices – that perpetuate structural violence and prevent access to healthcare services among young girls and women in Kisumu, Kenya. We purposively recruited young women (n = 36) enrolled in the Pamoja Community-Based Organization's DREAMS program in Kisumu, Kenya. Data were collected from the 2022–2023 cohort between June and July 2023 using semi-structured, in-depth interviews (n = 20) and two focused group discussions (n = 16). Guided by thematic analysis, data were analyzed in Atlas.Ti and organized into themes. This study found that community approaches such as safe spaces are instrumental in overcoming structural violence among young women by addressing three forms of barriers – institutional, sociocultural, and economic barriers – that limit HIV support service access. Institutional barriers encompassed distance and time to health facilities and provider attitude, while sociocultural barriers included knowledge gaps, stigma, cultural norms, beliefs, and practices, limiting health service access. Lastly, the socioeconomic barriers highlighted inadequate income, financial literacy, and financial dependency. Community safe spaces are vital for decreasing vulnerability and serve as critical points for accessing services and building

**Data availability statement:** All relevant data are within the paper.

**Funding:** The authors received no specific funding for this work.

**Competing interests:** The authors have declared that no competing interests exist.

capacity for young women. This is particularly important in rural areas where retrogressive societal norms create obstacles to obtaining essential health services. To effectively overcome structural violence, however, government support and a suitable policy environment are essential for implementing interventions to address the underlying root causes of structural violence and sustaining community-based safe spaces.

## Introduction

Over four decades of a sustained global fight against HIV have seen impressive gains made in the prevention and treatment of the epidemic [1]. The recent increase in the number of people with known HIV status, the number accessing anti-retrovirus therapy (ART), and the number virally suppressed have signaled the possibility of attaining the 95-95-95 global target for ending HIV [2]. One of the factors slowing the progress on the 95-95-95 target is the high HIV infections among girls and young women [2]. In 2017, for example, adolescent girls accounted for over 300,000 new infections weekly, increasing the global HIV burden [3]. With adolescent girls and young women accounting for more than 68% of HIV infections among the youth, the surge in new HIV infections in young women has remained high and is a pressing concern in sub-Saharan Africa [3].

Evidence suggests that structural violence, the indirect violence inherent in social institutions or structures, is a significant hindrance to HIV prevention, especially where inequities persist [4]. Because of its pervasiveness, structural violence is likely to be normalized, thus erasing history of health inequities [4,5]. Often facilitated by structural barriers - policies, norms, or practices that disadvantage marginalized groups, structural violence harms and prevents individuals from accessing basic needs and rights, hence limiting them from realizing maximum potentials in life [6]. For example, retrogressive sociocultural norms have contributed to increased gender-based violence (GBV), early marriages, and poor access to health and healthcare services [7–9]. Similarly, poverty and poor socio-economic status continue to facilitate transactional sex and promote intergenerational and multiple sexual relationships [10–12]. Further, some policies around sexual reproductive health, such as the restricted provision of contraceptives to adolescent girls in Kenya, have made it impossible for young women and girls to access these services [13]. Consequently, these barriers continue to inflict harm and increase the vulnerability of young women, further increasing the risk of HIV.

Though comprehensive HIV prevention strategies to address behavioral, biomedical, and structural factors that increase the risks of HIV infection among girls and young women exist, a substantial number of these interventions have been community-driven [14]. In Kenya, for example, self-organized groups, faith-based organizations, and self-help groups have been central to HIV prevention. They have contributed to the increased knowledge of HIV in the general population and increased demand for HIV care and treatment services [15]. Furthermore, community programs have positively contributed to the reduction of social barriers to HIV service uptake [16], including the reduction of stigma and discrimination [17] and the improvement of access to services and livelihoods, food, and water security [18].

Despite this significant role played by community-led initiatives in HIV prevention, inadequate evaluation frameworks have made it difficult to understand the role of community programs in overcoming structural violence, especially among rural young girls and women at risk. Therefore, this study aimed to evaluate the effectiveness of community-based safe spaces among young women receiving oral PrEP in overcoming structural barriers using the structural violence approach. As a framework, structural violence examines the systemic injustices and broader social and economic forces contributing to health disparities among marginalized populations. It is thus embedded in health inequities, living conditions, discrimination, and

economic exploitation [5]. We chose to examine the community safe spaces because of their unique approaches to health services, including providing biomedical services (HIV testing, oral PrEP, and contraceptives) and other structural interventions such as sociocultural and economic initiatives.

This study explored pathways by which community-based approaches and community-led programs have helped overcome structural violence to HIV prevention by examining the experiences of young women receiving PrEP through community safe spaces. Specifically, we examined the role and effectiveness of community-based safe spaces in overcoming systemic barriers to HIV prevention among young girls and women in a resource-constrained setting in Western Kenya.

## Methods

### Study location

This study was conducted in East Seme Ward, Seme sub-county in Kisumu, Kenya, where Pamoja Community-Based Organization (CBO) has implemented community HIV prevention interventions since 2007. Seme is about 25 kilometers from Kisumu City and borders Lake Victoria on the south. It is one of the poorest sub-counties in Kisumu County, with most residents involved in rain-fed subsistence farming, fishing, and/or motorcycle taxi business [19,20]. With poverty levels at 46%, Kisumu County is ranked one of the poorest counties in Kenya. The county also has the highest HIV prevalence, higher infectious disease burden, including tuberculosis and malaria, and high rates of teenage pregnancies [21,22].

### Study design

This cross-sectional study employed qualitative techniques, such as in-depth interviews (IDIs) and focus group discussions (FGD), to gather data from young women and girls participating in community HIV prevention programs within Pamoja CBO. The study involved community members who are part of the safe spaces program including peer mentors, mentees, and community advisory groups. The recorded interviews were transcribed and thematically analyzed, guided by the structural violence framework, and themes were allowed to emerge from the data.

### Community engagement

This study was conducted within the Pamoja Community-Based Organization (Pamoja CBO), a grassroots organization in Kisumu County, Kenya, working to empower communities to identify and address the most significant needs affecting their well-being. The organization has well-established community structures, including community advisory groups (CAGs) as links between the organization and the community. Before data collection, our research team met and worked with representatives from the CAGs to introduce the research and identify and seek permission to engage with the community safe spaces and their participants. In addition, the CAGs met regularly, were appraised of the research activities, and were allowed to ask any clarifying questions throughout the research period.

**Safe spaces defined.** In this study, we describe safe spaces as girls-only meeting avenues where they discuss issues affecting their well-being. A physical space is considered safe when it has been vetted by government officials, parents, guardians, and young women. Safe spaces consist of the mentor and the adoescent girls and young women (AGYW) as mentees and are designed to reduce power imbalances and enhance peer-to-peer communication and experience sharing among participants. The safe spaces are located centrally with the

communities to reduce transport costs that hinder attendance, participation, and service access. The physical safe spaces are primarily in community structures such as schools and churches. They are segmented into different demographic categories and vulnerabilities such as age, marital status, those with children, in school, or interventions such as PrEP. This segmentation allows for easy administration of specific services to each group. A mentor (usually one of their peers from the same geographical area) manages a safe space. Mentors undergo over five weeks of training on social asset building and work with other community resource persons, including community health promoters (CHPs) and other healthcare providers. They schedule activities in the safe spaces and ensure service providers are frequently invited to the safe spaces to talk and provide various services, including sexual and reproductive health services.

## Study participants

Data was collected between June and July 2023. We purposely recruited young women (n=36) enrolled in the Pamoja CBO's 2022-2023 DREAMS program for qualitative interviews and FGDs. Only beneficiaries belonging to safe spaces within the Seme sub-county were recruited to participate in IDIs or FGD. The goal was to explore the experiences of young women and evaluate the effectiveness of community-based safe spaces in mitigating the systemic barriers to HIV prevention, care, and management in young girls and women. Participants included women aged 18 to 24 who had been in the program for over six months. The six months duration was deemed sufficient for the project beneficiaries to receive essential and primary interventions while attending a safe space. These include evidence-based behavioral interventions (social asset-building skills), structural interventions (socioeconomic approaches), and biomedical interventions (PrEP, contraceptives, post-violence care).

## Ethical statement

This study received ethics approval from Amref Health (AMREF-ESRC P1396/2023). Prior to enrollment, we obtained written informed consent from all participants. Each participant was reimbursed up to KES 500 (USD 5) for transportation costs.

## Inclusivity in global research

Additional information regarding the ethical, cultural, and scientific considerations specific to inclusivity in global research is included in the Supporting Information (S1 Checklist).

## Qualitative data collection techniques

In-depth interviews and FGDs were conducted by trained qualitative interviewers well conversed in the local language. The interviews were conducted in English or the local language (Luo) per the participants' preferences and lasted about one and a half hours (60–90 minutes). To help contextualize the data, participants were asked to complete a participant log capturing sociodemographic characteristics, including age, education level, marital status, and number of parity (# of children) at the end of each interview.

**In-depth interview.** One-on-one in-depth interviews were conducted to clarify individual motivations, beliefs, feelings, and experiences with the safe spaces. In-depth interviews are critical to understanding individual experiences and allow researchers to explore sensitive topics with participants. An effective IDI involves creating a listening space where viewpoints are communicated [23]. We conducted IDIs with young girls and women (n = 20) participants of different ages to explore the effectiveness of community-driven safe spaces in addressing structural violence in the prevention and treatment of HIV in girls and young women.

In-depth interviews were conducted in individual homes by trained qualitative interviewers and lasted approximately 60 minutes. Questions during the interview included individual perceptions and experiences. For example, we asked participants questions like, "How does taking PrEP and other services at the safe space make you feel? What do you worry about when accessing safe spaces, and how would you describe your personal journey here?

**Focus-Group Discussion (FGD).** To qualitatively understand and describe the community norms, perceptions, and inter-group dynamics in the safe spaces, we conducted two focused group discussions with young women (n=16) participating in safe spaces from two villages. The FGDs were conducted in identified community spaces, which provided privacy and confidentiality to the participants and lasted for about 60-90 minutes. More broadly, participants were asked about various issues, including the socioeconomic benefits/barriers and the cultural norms and practices, especially those concerning how girls and young women are perceived and expected to behave. For example, participants were asked, "How do the families, partners, and spouses react to service provision in the safe space, especially for PrEP?" with specific probing on norms, traditions, family practices, and expectations for young women.

## Data analysis

All interviews were digitally recorded, transcribed verbatim, translated into English where applicable, and imported into Atlas. T.I. software for analysis. Transcripts were then iteratively analyzed deductively (using the structural violence framework) and inductively to develop a standard set of codes describing patterns observed in the data. In the first round of coding, the first author (PO) read all the transcripts and generated primary codes while assigning the generated codes to specific segments of the transcripts. In the second round, the second and the third authors (SO and WO) repeatedly went through the transcripts, merging and splitting codes where necessary. In the third and final round, all the researchers reviewed all the transcripts for emerging themes and categorized codes based on similarities of concepts and patterns within the data. Finally, analytic themes were identified using mixed techniques, including scrutiny and word-based techniques [24], starting from frequently used words and searching for subtle concepts in the data. Using the structural violence lens, we mapped out the key themes and the manifestations of structural violence addressed through the engagement of youth and young women in the community safe spaces (Fig 1).

## 3.0. Results

In this section, we present results on how community safe spaces overcome structural violence among young girls and women by supporting them in navigating various sociocultural, economic, and other structural barriers to healthcare access. We identified various barriers, including a) accessibility due to reduced physical distance and poor provider-client relationships (institutional barriers), b) cultural norms, beliefs, and practices (socio-cultural barriers), and c) inadequate PrEP and HIV knowledge, poor socioeconomic and livelihood challenges (socioeconomic barriers). In the subsequent section, we discuss these themes in detail, beginning with a description of our study participants, followed by a discussion of the analytic themes.

## Demographic characteristics

A total of 36 young women aged between 18 and 24 years (median age 20.0 and IQR 3 [19–22]) participated in the research study. A majority of the women had some primary school education (47.2%, n = 17), women with some high school education (42.0%, n = 15), and only (11.1%, n = 4) of the women had a college degree. Even though only 1 in 4 (25%, n = 9) of the women were married, 1 in 3 (33%, n = 12) had at least one child (Table 1).

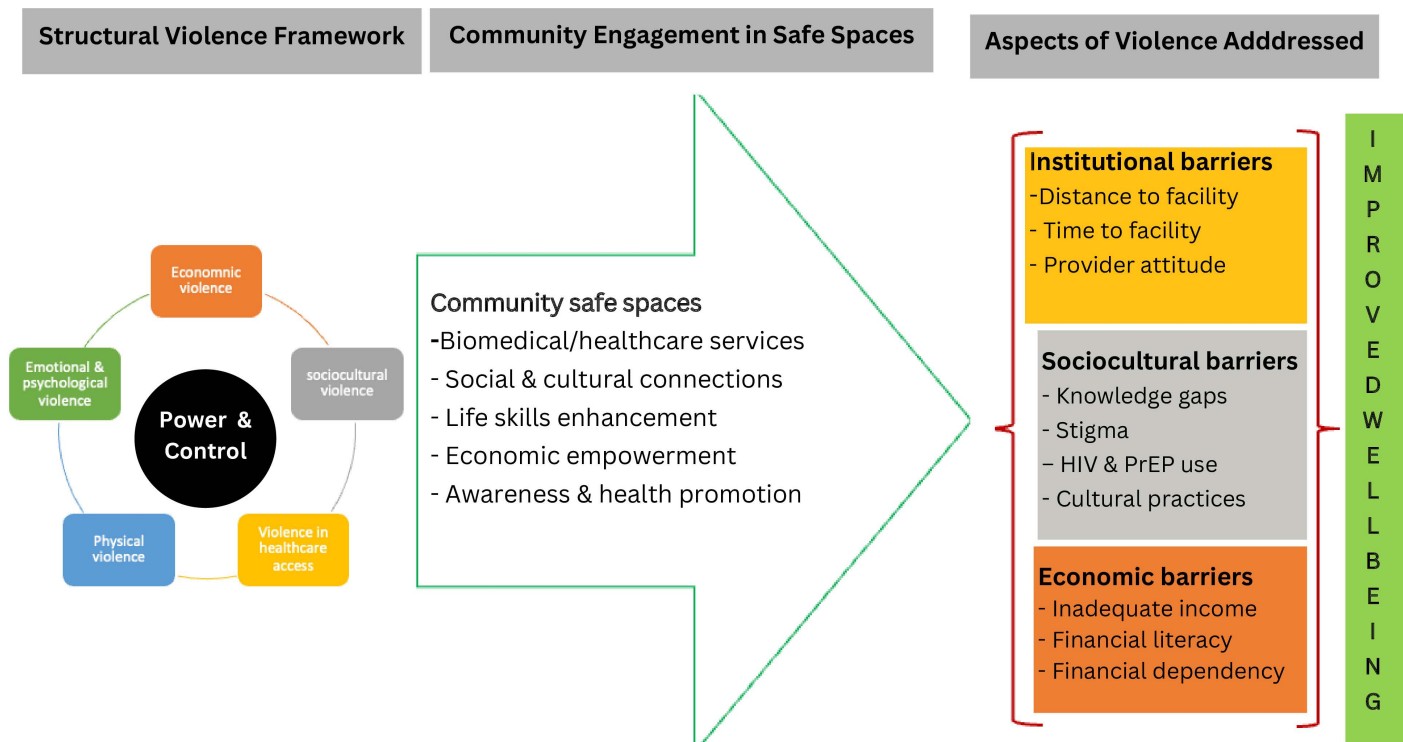

**Fig 1. A conceptual framework illustrating the role of community safe spaces in overcoming structural violence.**

**Table 1. Demographic characteristics of study participants.**

| Characteristics | Statistics (N = 36) |
| --- | --- |
| Age (Median) | 20.0 yrs. |
| IQR | 3.0 (19–22 yrs.) |
| Marital status (%) | |
| Married | 25.0 (n = 9) |
| Source of income % (n) | |
| SME | 83.3 (n = 30) |
| Education Level % (n) | |
| Some primary | 47.2 (n = 17) |
| Some high school | 42.0 (n = 15) |
| College | 11.1 (n = 4) |
| Religion (%) | |
| Christianity | 86.1 (n = 31) |
| Children % (n) | 33.3 (n = 12) |

SME: small and medium enterprises.

## Analytic themes

Three analytic themes—institutional barriers, sociocultural barriers, and economic barriers—emerged as the primary causes of structural violence addressed with the engagement of young women in community safe spaces (see Table 2). We categorized institutional barriers to encompass elements related to service access from healthcare facilities. This included

**Table 2. Analytic themes on structural barriers to PrEP uptake, related codes, and manifestation of structural violence.**

| Analytic theme | Primary code | Manifestation of structural violence |
|---|---|---|
| Institutional barriers | Access gap | Distance, time and privacy |
| | Provider attitude | Relationship, trust and support |
| Sociocultural barriers | Knowledge gap | Lack of awareness, and poor basic HIV knowledge |
| | Stigma and disclosure | Worry and fear of being labelled |
| | Cultural beliefs | Retrogressing gender roles and lack of decision-making powers |
| Economic barriers | Inadequate income | Lack of start-up capital |
| | Financial literacy | Poor financial skills business planning |
| | Financial dependency | Male sexual partners |

physical access factors (distance, travel time, affordability) and perceived healthcare providers' attitudes (rapport, trust, and support). We categorized sociocultural barriers as knowledge gap (awareness, health education), stigma and disclosure (worry, fear of being labeled), and cultural beliefs (retrogressive gender roles and power dynamics). Finally, we categorized the socioeconomic barriers to include inadequate income (lack of capital), financial literacy (poor financial literacy and business planning), and financial dependency (resorting to male sexual partners). We expand on these themes and illustrate how violence manifested itself due to these barriers and how attending safe spaces helped young women overcome these challenges.

## 3.1. Institutional barriers

**3.1.1. Improving physical access to health services through safe spaces.** This study revealed difficulties in accessing HIV-related health services due to the distance from where health facilities are located. For school-going girls, time and money to go to the health facility were always constraints since hospitals only operated during weekdays when they needed to be in school and were further away from home. Because safe spaces are located within the community, participants reported they could easily walk to them and get their services. Most participants reported safe spaces made it easy, especially to access their drugs (PrEP) and attend other scheduled check-up appointments. One participant, while narrating her challenges with access, said:

> "I am still going to school and cannot comfortably walk to the clinic to pick up my drugs (PrEP) and attend clinic appointments. Walking to and from the facility would mean I miss my classes and ask for permission from my teachers. However, for the sake of my drugs, I have to find a means. When this safe space started, the "doctor" allowed one of my group members to pick up my drugs for me at the safe space. I am happy that I only go to the hospital every three months and only when my check-up is due. (Focused group discussion, 02; participant 03)

A few participants expressed that the health facilities they visited lacked private rooms or areas to discuss their concerns. As a result, many missed their medication pickups or clinic appointments. During a detailed interview, one participant shared that on the days she needed to collect her medications, she would time her visit for late in the day when the hospital was less crowded, hoping to receive the nurse's attention more easily.

> "Let me tell you, going to that clinic has always been trouble for me. You go there, and there is only one open area where everything is handled. They call your name loudly and

*ask you sensitive questions so openly. It is embarrassing, and you can't discuss anything sensitive. I would go very late if I wanted to talk to the nurse. She is very lovely when she is not under pressure. Safe spaces have been a relief. I walk there when I want to, and I can have lengthy discussions with the nurse who visits. I think a lot has changed since I started attending Safe Space." (In-depth interview, participant 09)*

**3.1.2. Improving provider attitude and relations among young women.** Some participants shared their experiences with healthcare providers in public health hospitals and how the negative interactions made them fear going to the hospital unless it was the only alternative. Many mentioned resorting to seeking informal medical advice and treatment, including self-medication on suspected cases of sexually transmitted diseases. However, after joining Safe Spaces, some participants expressed confidence in the nurses and other health providers when they visit Safe Spaces.

*"My experience with the clinic has always not been good. I went there when I suspected I had contracted an STI from my abusive partner. Instead, the male clinician I got wanted to take advantage of me and started talking about how he could provide me with better sex and love. Safe space has been different; the health providers who visit our safe space are so friendly; they even visit us at home. They have helped me and other girls a lot." (In-depth interview, participant16)*

## 3.2. Sociocultural barriers

**3.2.1. Improving knowledge and awareness through Peer-to-peer education among girls and young women.** In the focus group discussion sessions, participants reported that most safe spaces conducted adherence education during their meetings. They described how lack of knowledge prevented some girls from enrolling in PrEP and how the education sessions improved their knowledge of non-adherence risks. In addition, the participants described how they felt supported to talk to their peers about the health benefits of PrEP. Some mentioned how other girls who did not want to take PrEP started to consider enrolling and improving their safe space numbers. This initiative greatly benefited most young women and girls who had started taking PrEP to understand the importance of adherence. A participant said:

*"In our safe space, we have nurses coming to teach us about adherence to PrEP; we share our journeys and educate those who are hesitant to be part of us. We do a lot of education; we refer those willing for screening. We are proud because our numbers have risen. We used to be only 10 girls and now we are twenty-five in this space a lot. As the other member mentioned, we are not afraid to talk to our partners about PrEP. Some of us are in discordant relationships, and this drug has saved their lives." (Focused group discussion 2, participant 09)*

Further, some participants mentioned how, initially, they were afraid of discussing their involvement in PrEP with their family members for fear of being labeled promiscuous. However, after several sessions in the safe space and learning from their peers, they felt comfortable talking to their family members and partners about PrEP.

*"The safe space has helped me gain courage and tell my family about my life and PrEP. I am not HIV positive, but it is shocking to know how HIV is regarded as a very serious illness, and testing positive, therefore, meant death. I felt my family would be worried if*

*they saw me taking the pills since they are like ARVs. Since I joined this safe space, I found information I did not know. Now, I am confident." (Focused group discussion 01, participant 12)*

In in-depth interview sessions, a participant mentioned how the fear of knowing her HIV status prevented her from enrolling in PrEP since this was a required step and HIV testing was primarily done in hospitals. However, when she started attending safe spaces, she got enough education and support to brave the test.

*"I know myself, and HIV testing has always scared me. So, I deliberately delayed starting PrEP even after I visited the safe space severally. I must confess that the way the nurse at the safe space taught me about HIV prevention and risk factors is what made me re-think. I knew that, with my kind of lifestyle, I had to do something. So, I decided to test during the testing session at the safe space. I got screened for PrEP eligibility, and now I am happily on PrEP" (In-depth interview, participant 05).*

**3.2.2. Overcoming stigma and disclosure among young women.** Most participants reported that fear of being labeled, talked about, and shunned from important community and social events initially prevented many young people from telling people, including their partners and families, that they had enrolled in PrEP. However, some participants noted that after many safe spaces started conducting adherence and disclosure education, most girls embraced PrEP and openly communicated their choices to enroll in PrEP to their partners, friends, and family members. One participant said:

*"In our safe space, we have education sessions conducted by the nurses and mentors. We have also been encouraged to share our personal experiences among ourselves. Now, we support each other on drug adherence. Personally, I have received support from my partner since I opened up and told him I am on PrEP. Also, as the other participant has mentioned, most of our partners have accepted testing for HIV." (Focused group discussion 01, participant 07)*

Some participants were worried about where they would pick the drugs because most healthcare workers were people known to them and their parents, and PrEP, according to them, was associated with HIV and promiscuity. Some said the hospitals were too public, and they needed private spaces to access drugs if they were ever to enroll. However, they felt comfortable starting PrEP after learning that they could go to the safe spaces and pick up their drugs.

*"PrEP in this community is like ARVs; it is difficult to pick the drugs if you go to the clinic and you find the people you know. They won't understand you are there for PrEP and not ARVs. Now, I am relieved because I go to a safe space to get everything I want." (In-depth interview, participant 20).*

**3.2.3. Overcoming cultural beliefs and practices.** In one focus group discussion, we found that community-led interventions have also helped deconstruct some cultural beliefs and practices that hindered access to care among young women. For example, one participant reported that being in a safe space has helped her participate in educational activities and learn information about sex and sexuality, which were not readily acceptable in her family and community, especially for girls. She narrates:

*"When I started coming to the safe space, my parents were not happy. They said girls are not supposed to go to such spaces on their own. Instead, they wanted me to stay home and help with household chores. They were furious when they heard that we were being taught about safe sex, and they threatened to beat me. I am happy I stayed because I have learned so many things that I did not know. I am also able to get so many services here instead of going to the hospital" (Focused group discussion 02, participant 05)*

Similarly, one participant recounted during an in-depth interview how she is consistently reminded of her identity as a girl, which comes with certain expectations. She described how her parents often dismissed her ideas, suggestions, and opinions, repeatedly emphasizing her gender role.

*"I live with my parents and three brothers and often feel neglected. Unlike my brothers, I don't have much of a voice. Frequently, I hear that I shouldn't do certain things as a girl. I recall when I wanted to enroll in a hairdressing class; I was told that my brothers were still in school and needed money for their fees. Last time, I felt hurt when my dad proposed the idea of marriage since I had finished high school. He mentioned that he deserved the bride price for sending me to school. It has really been difficult for me; my role is only to assist mom with all household chores. I can't say or suggest anything; yet, it is my family. But I thank God that after the mentor came and talked to my parents, they allowed me to start attending safe spaces. Though I am under strict supervision, including from my younger brothers, I must say that I have benefitted a lot from here and have started doing things independently." (In-depth interview, participant 15)*

### 3.3.0.  socioeconomic barriers

**3.3.1.  Livelihoods and economic empowerment.**  Most participants agreed that poverty, especially inadequate income and poor financial knowledge, was one of the reasons why some of them engaged in risky behaviors. However, after joining Safe Spaces and being trained in financial management and entrepreneurship skills, some reported improved income after starting small businesses, which helped them get the essential commodities they needed without relying on their parents or engaging in risky behaviors for financial gain.

*"I don't want to lie to you. There was a time when I had several men. I needed so much to get through high school. Sanitary pads, books, transport, my body oil, and pocket money. Some also contributed to my school after I had stayed home for so long that my parents could not raise the money. I wish we had safe spaces then. Now I have my small business, I am trained in financial literacy, and I can afford to buy the things I need. Honestly, my life, and the lives of others I know in this program, has really changed." (In-depth interview, 06)*

While sharing her experience in a focus group discussion, one participant confirmed that joining a safe space made her join others in economic activities to generate income. She observed that since then, she has been able to get her income to care for her menstrual needs. She illustrated:

*"I knew that to join a safe space, one had to be HIV positive. It took me a while to open up and ask questions. I felt ashamed of myself. I would go without sanitary pads even for two days and stay home. Since I joined, I have learned skills to help me generate money, and now I am more confident. I can ask my group members for sanitary pads and borrow money. We can no longer go without pads." (Focused group discussion, 05)*

Another participant in a focus group discussion shared her experience with economic vulnerabilities facing young women. She narrated how some of her friends who were young widows sought alternative activities, such as multiple sexual partners as benefactors to meet their needs. The participant narrated:

> "*The greatest challenge is that most of our members are young widows who rely mainly on farming. They struggle to feed their families during the dry season. So, they resort to seeking male sexual partners to meet their needs…. Since we joined Safe Spaces, at least we have been engaged in different income-generating economic activities . Some of us got business start-up kits and now have a stable income.*" (Focused group discussion 02, participant 08)

In summary, the community safe spaces have created opportunities for the young women and girls in Kisumu County, particularly Seme sub-county, to gain skills and enhance their capabilities to navigate systems of marginalization that increase their risk of acquiring HIV while also providing support to live positively with HIV.

## 4.0. Discussion

In this study, we sought to qualitatively explore how community safe spaces can help overcome structural violence to HIV prevention and treatment access among young women and girls living in Kisumu, Kenya. Our findings reveal that by continuously attending safe spaces, young women gained essential social skills, knowledge, and networks critical to navigating challenges to accessing PrEP and other health and wellbeing support services. We found that community safe spaces for HIV prevention services were instrumental in enhancing acceptance and uptake of PrEP through a) improving physical access, b) tackling cultural beliefs and practices, c) promoting peer-to-peer education, d) strengthening livelihoods and socioeconomic activities, and e) improving provider attitude and relations with young women.

Physical access emerged as a barrier for young women to access essential health services. More particularly, there were concerns of school-going young women who expressed challenges with distance and lack of time and money to travel to the hospital to get the needed services. We found that with the establishment of safe spaces, young women could access services during non-school days without worrying about money and distance. Because of the inability to access health facilities, many young women miss out on health education and services such as contraceptives and other prevention interventions. These findings expand on the literature on how vulnerable populations miss out on essential services because of poor accessibility to health institutions [25,26]. In low and middle-income countries, such as Kenya, poor access has been attributed to poor health-seeking behavior, including waiting until conditions worsen before getting care and patients defaulting from treatment regimens [27]. In rural areas, this is exacerbated by sparse health facilities, poor physical spaces with little to no privacy, and costly transport [27,28]. Efforts to mitigate barriers to physical access for HIV prevention have included outreach services and longer prescriptions to reduce the frequency of clinic appointments [29]. However, these efforts have faced challenges with sustainability and community involvement. Our study provides evidence for community-safe spaces as a way of enhancing community involvement and bridging the sustainability gap. Furthermore, we argue that integrating HIV prevention and treatment resources into community safe spaces can promote accessibility, acceptability, affordability, and service scalability.

Our study revealed that young women developed a negative attitude toward visiting public health facilities because of how poorly they were treated. However, they felt more

comfortable and established trust with healthcare providers in safe spaces and extended opportunities for home visits by health providers to strengthen their commitment to their health and behavior change. We extend the argument that health providers' knowledge and attitudes are critical to accessing health services [30,31]. and that the relationship that providers build with their clients is essential for the continuity of care and treatment outcomes [31,32]. Poor provider-patient relations and discriminatory practices remain challenging [33]. In low and middle-income countries, such as Kenya, these practices are responsible for defaulting from care from programs such as HIV treatment that require long-term engagement between healthcare providers and their clients [34,35]. Our study expands this argument by suggesting that safe spaces can help nurture healthcare providers' relationships with young women by building a trusted relationship. This can open ways to discuss sensitive health problems that young people struggle with and which are often neglected in the mainstream healthcare sector.

In this study, we found out that young women failed to access services due to inadequate knowledge of service availability and lack of awareness of the significance of healthcare. However, for the young women attending safe spaces, continuous peer-to-peer education offered by healthcare providers and peer mentors provided the needed information to overcome the knowledge barrier to health services. Health literacy – the ability to access, understand, and appropriately use health services information and take action – has been critical to the success of health interventions [36,37]. Poor or lack of health literacy is associated with poor health outcomes and higher morbidity and mortality, especially in chronic conditions such as HIV [36,38,39]. Thus, allowing young people to participate in community programs can increase awareness and improve health outcomes [40]. According to our study findings, young people's knowledge, attitudes, and perception of seeking help also improve when participating in community programs. Collaborative learning, a common strategy in community programs, provides opportunities for knowledge sharing, especially among young people, and thus plays a significant role in promoting HIV and reproductive health literacy [41,42].

Stigma continues to be a significant obstacle to accessing health services. For young women on PrEP, stigma is linked to perceptions of being HIV-positive or sexually promiscuous making its uptake and use challenging to accept among young people [43]. Strategies such as social support – the support that is accessible to individuals through social networks – have been identified as effective in reducing stigma and are closely linked to positive health outcomes [44,45]. In HIV management, social support has facilitated greater adherence to HIV treatment. For example, Kibaara et al. (2016) have shown that HIV-positive individuals in Kenya with treatment buddies had higher levels of adherence than those without [46]. Closely linked to stigma in HIV treatment and prevention is non-disclosure, which has been argued to result from fear of being stigmatized and labeled. Like stigma, lack of disclosure in PrEP is perceived to stem from the fear of being labeled HIV-positive and sexually immoral and requires strong social support to overcome [47,48]. Our study expands on this literature and identifies community safe spaces as potential avenues for reducing stigma related to PrEP and encouraging disclosure through enhanced social support.

Despite extensive awareness creation on retrogressive societal norms and increased focus on initiatives supporting the girlchild in this region [49,50], our study found sex and sexuality are still difficult topics for many rural families and that girls and young women still face restrictions on how and what they should learn. Our study reveals that gender norms continue to limit the potential of many young women since families expect them to stay home and do household chores. Venturing out either to seek information or openly talk about their womanhood is considered taboo and might result in punishment. Additionally, economic

vulnerability remains a threat to the health and wellbeing of young girls and women in resource-constrained societies. Our study revealed that young women felt more vulnerable when they did not have adequate income to cover their basic needs, consequently falling victim to transactional sex and multiple sexual partners. However, participating in safe spaces equipped them with financial and entrepreneurial skills to start businesses, thus establishing some financial security. Existing body of literature that has linked economic empowerment to improved household income, food security, and quality of health [51,52]. Economic empowerment – individuals' capacity to participate in development initiatives, decision-making, and negotiation for the fair distribution of resources – can enhance young women's access to resources and opportunities, including employment, financial access, training, and information.

Evidence from this study contributes to the body of literature that has highlighted how harmful gender norms continue to create obstacles to healthcare services [19,53,54]. In sub-Saharan Africa, social institutions such as the family and religion have contributed to negative health-seeking behavior and prevented the uptake of critical health interventions [55–57]. For example, cultural practices such as family-organized early marriages and polygamy are but a few retrogressive cultural practices that undermine the potential of young women [58–60]. In addition, gender roles in sub-Saharan Africa have disproportionately affected women, especially girls and young women, who bear the burden of household chores compared to their male counterparts [61–63]. Thus, by overly being responsible for chores such as water fetching, young women are left behind from participating in behavioral interventions and meaningful life skills-building programs, potentially leaving them more vulnerable and at risk.

These findings contribute to the existing body of literature that has linked economic empowerment to improved household income, food security, and quality of health [51,52]. Economic empowerment – individuals' capacity to participate in development initiatives, decision-making, and negotiation for the fair distribution of resources – can enhance young women's access to resources and opportunities, including employment, financial access, training, and information. However, the establishment and strengthening of the socioeconomic status among HIV-affected families have been a preserve of community partners [64]. This study suggests that community safe spaces are promising avenues for introducing socioeconomic interventions for young women because of the peer support and trust that safe spaces instill in them. Our study suggests that community safe spaces are more trusted and can offer avenues for health education, empower young women, enhance the uptake of health services, and improve health and well-being.

## Limitations

Despite the strengths in the rigor of data collection, the timeliness of this study, and intriguing results, we acknowledge the potential weaknesses of the paper. First, the cross-sectional and qualitative nature of this study limits its generalizability. Additionally, the study participants were drawn mainly from the DREAMS Program implemented by Pamoja CBO in the western region of Kenya; hence, the results may only be inferred to similar places or programs. Secondly, though DREAMS programs have been implemented across different ages and segmented by age and vulnerability, our study focused on young women (18-24 yrs.) without segmenting participants into any category, thus limiting age-specific experiences. The findings from this work do not reflect the experiences of alternative genders of the same age living in the same context. The findings from this work are, however, comparable to previous studies that have explored vulnerabilities to HIV by gender and age [65].

## 5.0. Conclusion

Structural barriers to HIV prevention continue to harm disadvantaged populations, especially young women. Community programs such as those that target literacy levels, social support, economic empowerment, and the use of community resource persons can dismantle these structural barriers. However, assessing successful interventions for sustainability will be critical. It is essential that community approaches, e.g., community safe spaces, are documented, evaluated, and communicated to communities that can benefit from them. By bringing these insights to the attention of HIV/AIDS practitioners, implementers, and the community, we can harness the power of community approaches to overcome structural violence and improve HIV treatment outcomes.

## Supporting information

**S1 Appendix. In-depth interview guide.**
(DOCX)

**S2 Appendix. Raw dataset for demographic characteristics of young women enrolled in the study.**
(DOCX)

**S1 Checklist. Inclusivity in global research.**
(DOCX)

## Author contributions

**Conceptualization:** Patrick Mbullo Owuor, Elizabeth Opiyo Onyango.

**Data curation:** Patrick Mbullo Owuor, Silvia Achieng Odhiambo, Wicklife Odhiambo Orero.

**Formal analysis:** Patrick Mbullo Owuor, Judith Atieno Owuor, Elizabeth Opiyo Onyango.

**Methodology:** Patrick Mbullo Owuor, Silvia Achieng Odhiambo, Wicklife Odhiambo Orero, Judith Atieno Owuor, Elizabeth Opiyo Onyango.

**Project administration:** Patrick Mbullo Owuor, Silvia Achieng Odhiambo.

**Supervision:** Patrick Mbullo Owuor, Judith Atieno Owuor, Elizabeth Opiyo Onyango.

**Writing – original draft:** Patrick Mbullo Owuor.

**Writing – review & editing:** Patrick Mbullo Owuor, Silvia Achieng Odhiambo, Wicklife Odhiambo Orero, Judith Atieno Owuor, Elizabeth Opiyo Onyango.

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
