## [Decision Letter · Decision Letter 0]

6 May 2024

PGPH-D-24-00463

Overcoming structural violence through community-based approaches: Insights from community-based HIV prevention programs in Kisumu, Kenya.

Dear Dr. Owuor,

Thank you for submitting your manuscript to PLOS Global Public Health. After careful consideration, we feel that it has merit but does not fully meet PLOS Global Public Health’s publication criteria as it currently stands. Therefore, we invite you to submit a revised version of the manuscript that addresses the points raised during the review process.

We look forward to receiving your revised manuscript.

Kind regards,

Jianhong Zhou

Staff Editor

Journal Requirements:

2. Please include a title page at the beginning of your manuscript file that lists full author names and institute addresses. This should not be uploaded as a separate file.  

4. In the online submission form, you indicated that "Data will be available upon request". All PLOS journals now require all data underlying the findings described in their manuscript to be freely available to other researchers, either 1. In a public repository, 2. Within the manuscript itself, or 3. Uploaded as supplementary information.

Additional Editor Comments (if provided):

Reviewers' comments:

Reviewer's Responses to Questions

**Comments to the Author**

1. Does this manuscript meet PLOS Global Public Health’s publication criteria ? Is the manuscript technically sound, and do the data support the conclusions? The manuscript must describe methodologically and ethically rigorous research with conclusions that are appropriately drawn based on the data presented.

Reviewer #1: Partly

2. Has the statistical analysis been performed appropriately and rigorously?

Reviewer #1: N/A

3. Have the authors made all data underlying the findings in their manuscript fully available (please refer to the Data Availability Statement at the start of the manuscript PDF file)?

Reviewer #1: No

4. Is the manuscript presented in an intelligible fashion and written in standard English?

Reviewer #1: Yes

5. Review Comments to the Author

Reviewer #1: Thank you for the opportunity to review this manuscript. It's a qualitative paper describing client perspectives on a community based safe space HIV prevention programme and effect/ impact on access to PrEP, adherence to PrEP, stigma and task shifting. The problem with the manuscript is that it's not very clear in its objectives and the descriptions of the interviews generalized as opposed to being specific and some conclusions not supported by the evidence presented. I have listed below specific concerns/ issues I had with the manuscript

- Title: the title refers to community HIV prevention programmes when in reality the authors studied one type of community-based programme in one location i.e. Safe spaces programme in Kisimu

Abstract

- The abstract introduction mentions that HIV prevalence in Kenya decreased to 4.9%. Over what time and from what % baseline

- The abstract introduction is not speaking to the title of the paper. I was expecting to read about structural violence among AGYW. At minimum, I expected the abstract introduction to mention structural violence and its relationship to HIV acquisition and lack of access to HIV care and treatment

- the abstract methods doesn't mention what community-based HIV prevention programmes were explored and what informed them. Was there a framework used for the exploration

- the conclusion is not speaking to the results presented

Introduction

page 8, lines 8- 11: The issue of HIV incidence among AGYW is more nuanced than this. In SSA, incidence in this group has been going down for many years although it remains highest compared to other risk groups. So, I wouldn't say increased, rather high

Methods

page 11, line 110- 114: Why only safe spaces when DREAMS was implemented in many other community -led interventions

Page 12, line 117: provide the equivalent amount in US dollars

Page 12, line 138- 142: the authors list the objectives of the in-depth interviews as to explore successes , failures and challenges of PaMoja's community-based HIV prevention and community programmes I think this should be specific to their safe spaces programme since you only interviewed young women attending the safe spaces programme.

Page 12, line 142: please include the interview guide used during the indepth interviews

Page 12, line 144: see comment above regarding HIV prevention programmes when you only spoke to people in the safe spaces programme

Results

Page 13, line 157: Please present the median and IQR for age

Page 13, line 160: also include the proportion of the young women who were HIV positive and on ART

Page 13, Line 163: the authors stated that the study revealed difficulties in accessing health services. Was this specific for HIV related services

Page 14, line 192: this sub-title doesn't reflect what is written below. perhaps add something about adherence

Page 15, line 224: It seems the clients confused ARVs with PrEP

Page 17, line 283: what sort of needs of the young wonen were met by the mentors

Discussion

Page 21, line 405: I am not convinced about this task shifting story

Page 22, line 421: I think just be specific to HIV prevention as was outlined in the introduction. HIV management covers care and treatment as well

6. PLOS authors have the option to publish the peer review history of their article (what does this mean? ). If published, this will include your full peer review and any attached files.

**Do you want your identity to be public for this peer review?** For information about this choice, including consent withdrawal, please see our Privacy Policy .

Reviewer #1: No

---

## [Decision Letter · Decision Letter 1]

13 Aug 2024

PGPH-D-24-00463R1

Overcoming structural violence through community-based safe-space for HIV prevention: Insights from Kisumu, Kenya

Dear Dr. Owuor,

Thank you for submitting your manuscript to PLOS Global Public Health. After careful consideration, we feel that it has merit but does not fully meet PLOS Global Public Health’s publication criteria as it currently stands. Therefore, we invite you to submit a revised version of the manuscript that addresses the points raised during the review process.

Please address the additional suggestions made by the reviewer.

We look forward to receiving your revised manuscript.

Kind regards,

Avanti Dey, PHD

Staff Editor

Journal Requirements:

Additional Editor Comments (if provided):

Reviewers' comments:

Reviewer's Responses to Questions

**Comments to the Author**

1. If the authors have adequately addressed your comments raised in a previous round of review and you feel that this manuscript is now acceptable for publication, you may indicate that here to bypass the “Comments to the Author” section, enter your conflict of interest statement in the “Confidential to Editor” section, and submit your "Accept" recommendation.

Reviewer #1: All comments have been addressed

2. Does this manuscript meet PLOS Global Public Health’s publication criteria ? Is the manuscript technically sound, and do the data support the conclusions? The manuscript must describe methodologically and ethically rigorous research with conclusions that are appropriately drawn based on the data presented.

Reviewer #1: Partly

3. Has the statistical analysis been performed appropriately and rigorously?

Reviewer #1: N/A

4. Have the authors made all data underlying the findings in their manuscript fully available (please refer to the Data Availability Statement at the start of the manuscript PDF file)?

Reviewer #1: Yes

5. Is the manuscript presented in an intelligible fashion and written in standard English?

Reviewer #1: Yes

6. Review Comments to the Author

Reviewer #1: Thank you for the opportunity to review this revised manuscript. The authors did a good job of addressing the comments I had on the manuscript. I however still have some more comments based on the information provided

Abstract

Introduction: I thought the authors just looked at HIV prevention, specifically a community led, safe spaces PrEP program. So, I would recommend removing the care and management from the objectives

Methods: I would make the abstract introduction more succinct and add details about how the study was actually conducted to the abstract methods e.g. study location, study population etc.

Results: the authors refer to barriers to structural violence in the abstract results section, this doesn't make sense. I thought the idea was to overcome structural violence and not the barriers to structural violence. When you overcome barriers to structural violence you facilitate the structural violence

Introduction

Line 60- 61: the authors wrote about eroding progress. Maybe use impeding progress or slowing the progress. There is progress still being made

Line 103: the authors wrote about successes not recognizing CBOs; successes themselves don't recognize CBOs but can lead to greater recognition of the CBOs

Line 108: structural barriers to what?

Line 135- 136: the authors refer to structural barriers that entrench structural violence within HIV management. shouldn't this be HIV prevention rather

Line 137- 138: care and management should be replaced by HIV prevention

Methods

Line 147: which lake?

Line 207: please refer to HIV prevention as opposed to HIV care and treatment

Results

Line 221: please include the ages included in the IQR i.e. 19- 22 as opposed to 3

Line 231: remove the word also

Line 275: disclosure on taking PrEP? I didn't know that there was an issue with disclosure when it comes to taking PrEP. Please elaborate

Discussion

line 375: Please remove the phrase community approaches such as. The authors only looked at one approach

Line 376: I would remove the phrase barriers to structural violence. See earlier comment

Line 379: I am not convinced about this disclosure story

Line 433- 446: since the safe space under consideration focused on PrEP, this is not relevant

Line 461- 464: again, we are talking PrEP, so irrelevant

Conclusion

Line 512-513: structural barriers to PrEP access?

Line 514: shouldn't HIV management be HIV prevention

7. PLOS authors have the option to publish the peer review history of their article (what does this mean? ). If published, this will include your full peer review and any attached files.

**Do you want your identity to be public for this peer review?** For information about this choice, including consent withdrawal, please see our Privacy Policy .

Reviewer #1: No

---

## [Decision Letter · Decision Letter 2]

27 Sep 2024

PGPH-D-24-00463R2

Overcoming structural violence through community-based safe-space for HIV prevention: Insights from Kisumu, Kenya

Dear Dr. Owuor,

Thank you for submitting your manuscript to PLOS Global Public Health. After careful consideration, we feel that it has merit but does not fully meet PLOS Global Public Health’s publication criteria as it currently stands. Therefore, we invite you to submit a revised version of the manuscript that addresses the points raised during the review process.

We look forward to receiving your revised manuscript.

Kind regards,

Annesha Sil, Ph.D.

Staff Editor

PLOS 

Journal Requirements:

Additional Editor Comments (if provided):

Reviewers' comments:

Reviewer's Responses to Questions

**Comments to the Author**

1. If the authors have adequately addressed your comments raised in a previous round of review and you feel that this manuscript is now acceptable for publication, you may indicate that here to bypass the “Comments to the Author” section, enter your conflict of interest statement in the “Confidential to Editor” section, and submit your "Accept" recommendation.

Reviewer #1: All comments have been addressed

Reviewer #2: (No Response)

2. Does this manuscript meet PLOS Global Public Health’s publication criteria ? Is the manuscript technically sound, and do the data support the conclusions? The manuscript must describe methodologically and ethically rigorous research with conclusions that are appropriately drawn based on the data presented.

Reviewer #1: Yes

Reviewer #2: Partly

3. Has the statistical analysis been performed appropriately and rigorously?

Reviewer #1: N/A

Reviewer #2: N/A

4. Have the authors made all data underlying the findings in their manuscript fully available (please refer to the Data Availability Statement at the start of the manuscript PDF file)?

Reviewer #1: Yes

Reviewer #2: Yes

5. Is the manuscript presented in an intelligible fashion and written in standard English?

Reviewer #1: Yes

Reviewer #2: Yes

6. Review Comments to the Author

Reviewer #1: Thank you for the opportunity to re-review this manuscript. The authors have addressed all the comments I had. I have no further comments

Reviewer #2: Thank you for the opportunity to review this paper on overcoming structural violence through Safe Spaces in Kenya. This is an important topic, however I think there are some major flaws with how the study is at least described. For example, while I do believe in the concept of structural violence, but right now it’s serving more as a distraction to your argument and the tightness of your paper because it’s not consistently described or framed that way and the results also aren’t framed in that way leading to some conceptual confusion. Also, there are many missing details of the methods so that one cannot adequately assess the quality. The research is described a participatory, but nothing in the methods suggests that is true. And contradictory approaches are described to analysis. I also find the results lacking in a structure that carries the reader through in a logical way and the discussion is not always closely aligned with the results. Additional detailed feedback is available below to help the authors think through these issues further.

• Abstract

o The intro talks about structural barriers, but not structural violence so it doesn’t seem aligned with the article title. The reader needs to understand this link between structural barriers and then what you refer to as structural violence (also in the results).

o Why is the RE-AIM theory relevant to this inquiry?

o The way the results are framed, I’m not clear that all the themes came from the young women.

• Introduction

o I see that you do explain your use of structural violence in the introduction, but it comes after you first mention structural violence (line 113) and first mention the study aim. The reader needs to understand your use of this terminology before you mention the aim. I would adjust the introduction to do so. I think you can also do it in less than 2 paragraphs and without as much explanation of only Paul Farmer, but rather introduce that concept, where it came from, what it means, and tie to what we know as structural violence in the AGYW HIV healthcare space.

• Methods

o You’re missing references from your study location section.

o I don’t really understand your use of the RE-AIM framework in this participatory qualitative work. That framework feels more useful when doing a proper program evaluation. Also, it feels like it takes away from the described value in community-led knowledge. I actually rather wonder if the concept of structural violence is what you should explain as your study conceptual grounding.

o Typo with your sentence on lines 177 and 178 – double use of ‘program’ – “program beneficiaries of this program”

o Explain why the threshold of 6-months of participation was deemed meaningful.

o In the Safe Spaces description you might want to mention the training and support a mentor receives to lead these groups.

o Qualitative data collection

Who conducted the IDIs? And I’m unclear why the IDI covered ‘group strategies’ as you mention in line 209. The value of an IDI to cover individual experiences.

You don’t sufficiently describe why you conducted FGDs, drawing on the value of FGDs, which is to unpack community level norms, attitudes, experiences. And who conducted those? And how did you decide which participants were in the IDIs vs. FGDs? And were the two groups segmented somehow, similar to how you have varying ages for the IDIs?

Where were the IDIs and FGDs done?

This section doesn’t adequately describe the structured areas of questioning in both data collection formats.

o Analysis

How many people and who was involved in analysis?

How was ICR done procedurally?

How could the analysis be guided by both grounded theory the RE-AIM framework, especially as grounded theory means you let theories emerge from the data? And how was the RE-AIM framework applied? If you did you use, what is the risk in doing so (to be included in the discussion)?

What was done after coding to analyze the data and come to your interpretations?

o I’m not seeing anything in your methods to justify calling it participatory action research.

• Results

o Did you conduct a survey with the qual participants to capture these descriptive characteristics? If so, that should be described in your method.

o What’s a sharing session (line 252)? And I’m not sure the quote exemplifies how the program deconstructed cultural beliefs as it doesn’t speak to prior beliefs.

o The quote around confidence and disclosure is important, but I’m not sure you’re framing in terms of structural barriers (lines 280-289).

o Lines 291-301, while important, don’t demonstrate how Safe Spaces overcame stigma. This is more setting the stage for stigma experiences. You could start your results with described structural barriers, including stigma, and then have sections on the role of Safe Spaces in overcoming them. I do feel there needs to be more an understanding of their perceived structural barriers first.

o Line 335 – I would remove the HIV positive part of the quote as it doesn’t lend itself to understanding the rest of the quote.

o Similar comment as above that lines 342-351 is more about the structural barriers and not about the role of Safe Spaces in navigating them.

o Overall, I feel the results could be restructured to reflect first the barriers and then how Safe Spaces were perceived as addressing them.

• Discussion

o The first paragraph of the discussion should summarize the key findings, not just the study approach. Combine the first and second paragraphs.

o I didn’t see in the results anything to support the literacy issue or clearly the task shifting. I also am not sure I saw the women talk about their improved health and well-being or improved relationships with service providers.

o I don’t think the health literacy finding as presented in the results is strong enough to warrant two paragraphs about it in the discussion.

o Overall, I wouldn’t put sub-headings in the discussion and would try to tighten it. Also you should weave reference to your findings into your review and comment of other topics and suggestions. For example with the social support, call attention to what was seen in the data, and then use other literature to describe its importance and future recommendations. You do this a little in the economic empowerment, but your last sentence there is a little weak as I’m not sure what it’s suggesting or recommending as a learning drawing from your work.

o Line 495, you say you were focused on barriers to PrEP access, which wasn’t the framing earlier.

7. PLOS authors have the option to publish the peer review history of their article (what does this mean? ). If published, this will include your full peer review and any attached files.

**Do you want your identity to be public for this peer review?** For information about this choice, including consent withdrawal, please see our Privacy Policy .

Reviewer #1: No

Reviewer #2: No

---

## [Decision Letter · Decision Letter 3]

22 Nov 2024

PGPH-D-24-00463R3

Overcoming structural violence through community-based safe- spaces: Qualitative insights from young women on PrEP in Kisumu, Kenya.

Dear Dr. Owuor,

Thank you for submitting your manuscript to PLOS Global Public Health. After careful consideration, we feel that it has merit but does not fully meet PLOS Global Public Health’s publication criteria as it currently stands. Therefore, we invite you to submit a revised version of the manuscript that addresses the points raised during the review process.

We look forward to receiving your revised manuscript.

Kind regards,

Nancy Angeline Gnanaselvam

Academic Editor

Journal Requirements:

Additional Editor Comments (if provided):

Kindly address the comments of the reviewer.

Reviewers' comments:

Reviewer's Responses to Questions

**Comments to the Author**

1. If the authors have adequately addressed your comments raised in a previous round of review and you feel that this manuscript is now acceptable for publication, you may indicate that here to bypass the “Comments to the Author” section, enter your conflict of interest statement in the “Confidential to Editor” section, and submit your "Accept" recommendation.

Reviewer #1: All comments have been addressed

Reviewer #2: (No Response)

2. Does this manuscript meet PLOS Global Public Health’s publication criteria ? Is the manuscript technically sound, and do the data support the conclusions? The manuscript must describe methodologically and ethically rigorous research with conclusions that are appropriately drawn based on the data presented.

Reviewer #1: Yes

Reviewer #2: Partly

3. Has the statistical analysis been performed appropriately and rigorously?

Reviewer #1: N/A

Reviewer #2: N/A

4. Have the authors made all data underlying the findings in their manuscript fully available (please refer to the Data Availability Statement at the start of the manuscript PDF file)?

Reviewer #1: No

Reviewer #2: Yes

5. Is the manuscript presented in an intelligible fashion and written in standard English?

Reviewer #1: Yes

Reviewer #2: Yes

6. Review Comments to the Author

Reviewer #1: Thank you for the opportunity to re-review this manuscript. I did not have comments on the previous version. I also went through reviewer 2 comments and think the authors addressed those comments well.

Reviewer #2: Thank you for the opportunity to re-review this paper on a qualitative evaluation of safe spaces and its role in overcoming structural violence among AGYW. The paper is much improved, however I still had some concerns about the description of the methods as participatory research and also with the strength of the results in how they are presented. My comments are as follows:

• Methods

o I wouldn’t classify having a CAB as participatory research – it’s community engaged, but not participatory unless they are setting the research agenda, informing the methods, involved in data collection and analysis

• Results

o For the first two themes in results, the presentation of a single quote with one paragraph of text doesn’t lend me to have a sense that these were dominant themes. They feel inadequately supported. Can you give a better sense of their prevalence, so to speak, and some more richness to their presentation? Some of the richness and nuance actually comes through in the section on peer-to-peer education so perhaps you can reorganize. You could describe them as intersecting themes so not try to divide them into three sections and then give description and examples of how they intersect.

You might move the provider attitude to a section on healthcare access barriers that are overcome as its related. It could start with a framing of the barriers to healthcare access from a geographic and provider perspective. The provider attitude section currently also feels underdeveloped to stand as a strong theme.

o For the livelihoods theme, the idea of transactional sex comes out of nowhere as it is not mentioned in the introduction and there is nothing beforehand from the qual about it being an issue or connected to structural violence. This could be resolved a bit by framing the start of this paragraph differently and raising transactional sex as a theme among the girls, which was described as being alleviated…

o Generally I think the results feel underdeveloped in a number of areas and may also be strengthened by some re-organization. Is there a way the concept of structural violence could help with that? Does it offer different categories that these could then be ordered in, for example?

• Discussion

o For the stigma paragraph, you may want to review and cite growing literature on PrEP stigma among AGYW. It’s not new that PrEP is associated with ARVs and HIV stigma. See for example:

Hartmann, M., Nyblade, L., Otticha, S., Marton, T., Agot, K., & Roberts, S. T. (2024). The development of a conceptual framework on PrEP stigma among adolescent girls and young women in sub‐Saharan Africa. Journal of the International AIDS Society, 27(2), e26213.

7. PLOS authors have the option to publish the peer review history of their article (what does this mean? ). If published, this will include your full peer review and any attached files.

**Do you want your identity to be public for this peer review?** For information about this choice, including consent withdrawal, please see our Privacy Policy .

Reviewer #1: No

Reviewer #2: No

---

## [Editor Report · Decision Letter 4]

16 Dec 2024

PGPH-D-24-00463R4

Overcoming structural violence through community-based safe- spaces: Qualitative insights from young women on PrEP in Kisumu, Kenya.

Dear Dr. Owuor,

Thank you for submitting your manuscript to PLOS Global Public Health. After careful consideration, we feel that it has merit but does not fully meet PLOS Global Public Health’s publication criteria as it currently stands. Therefore, we invite you to submit a revised version of the manuscript that addresses the points raised during the review process.

We look forward to receiving your revised manuscript.

Kind regards,

Nancy Angeline Gnanaselvam

Academic Editor

Journal Requirements:

Additional Editor Comments (if provided):

     1. Conceptual framework diagram is required to be presented in the article to establish the relationship between the themes and variables
---

## [Editor Report · Decision Letter 5]

9 Jan 2025

Overcoming structural violence through community-based safe- spaces: Qualitative insights from young women on PrEP in Kisumu, Kenya.

PGPH-D-24-00463R5

Dear Dr. Owuor,

We are pleased to inform you that your manuscript 'Overcoming structural violence through community-based safe- spaces: Qualitative insights from young women on PrEP in Kisumu, Kenya.' has been provisionally accepted for publication in PLOS Global Public Health.

Best regards,

Nancy Angeline Gnanaselvam

Academic Editor

Authors' response to my comments are accepted.